# The Diagnosis of Fungal Neglected Tropical Diseases (Fungal NTDs) and the Role of Investigation and Laboratory Tests: An Expert Consensus Report

**DOI:** 10.3390/tropicalmed4040122

**Published:** 2019-09-24

**Authors:** Roderick Hay, David W Denning, Alexandro Bonifaz, Flavio Queiroz-Telles, Karlyn Beer, Beatriz Bustamante, Arunaloke Chakrabarti, Maria de Guadalupe Chavez-Lopez, Tom Chiller, Muriel Cornet, Roberto Estrada, Guadalupe Estrada-Chavez, Ahmed Fahal, Beatriz L Gomez, Ruoyu Li, Yesholata Mahabeer, Anisa Mosam, Lala Soavina Ramarozatovo, Mala Rakoto Andrianarivelo, Fahafahantsoa Rapelanoro Rabenja, Wendy van de Sande, Eduard E Zijlstra

**Affiliations:** 1The International Foundation for Dermatology, London W1T 5HQ, UK; 2The Global Fund for Fungal Infections, 1208 Geneva, Switzerland, and the University of Manchester, Manchester M13 9PL, UK; 3Hospital General de México, “Dr. Eduardo Liceaga”, CP 06720, Mexico; a_bonifaz@yahoo.com.mx; 4Department of Public Health, Hospital de Clinicas, Federal University of Parana, 80060-900 Curriba, Parana, Brazil; queiroz.telles@uol.com.br; 5Centers for Disease Control and Prevention, Atlanta, GA 30329, USA; ydh7@cdc.gov (K.B.); tnc3@cdc.gov (T.C.); 6Tropical Medicine, Infectious Diseases and Dermatology Department at the Hospital Cayetano Heredia, Lima 15102, Peru; ana.bustamante@upch.pe; 7Postgraduate Institute of Medical Education and Research, Chandigarh 160012, India; arunaloke@hotmail.com; 8Hospital General de Acapulco, Secretaria de Salud Guerrero, C.P. 39901, Mexico; chavezg13@live.com.mx; 9Laboratoire de Parasitologie-Mycologie, Grenoble Alpes University, CNRS, Grenoble INP, CHU Grenoble Alpes, F-38000, France; mcornet@chu-grenoble.fr; 10Community Dermatology Mexico C.A., Acapulco 39850, Guerrero, Mexico; restrada_13@hotmail.com (R.E.);; 11The Mycetoma Research Centre, Khartoum, Soba University Hospital, P.O. Box 102, Sudan; ahfahal@hotmail.com; 12School of Medicine and Health Sciences, Universidad del Rosario, Bogota 111211, Colombia; beatrizlgomez@hotmail.com; 13Peking University First Hospital, Research Centre for Medical Mycology, Peking University, Beijing 100034, China; mycolab@126.com; 14Department of Medical Microbiology, National Health Laboratory Services and School of Laboratory Medicine and Medical Sciences, Nelson R. Mandela School of Medicine, University of KwaZulu-Natal, Berea, Durban 4001, South Africa; MAHAB@ukzn.ac.za; 15Nelson R Mandela School of Medicine, University of Kwazulu Natal, Berea, Durban 4001, South Africa; mosama@ukzn.ac.za; 16Faculty of Médicine, Antananarivo B.P 375 - 101, Madagascar; lsramarozatovo@gmail.com (L.S.R.); frapelanoro@yahoo.fr (F.R.R.); 17Centre d’Infectiologie Charles Mérieux, Université d’Antananarivo, Antananarivo BP 4299, Madagascar; mala@cicm-madagascar.com; 18Erasmus MC, Department of Medical Microbiology and Infectious Diseases, 3000 CA Rotterdam, The Netherlands; w.vandesande@erasmusmc.nl; 19Drugs for Neglected Disease Initiative, 1202 Geneva, Switzerland; ezijlstra@dndi.org

**Keywords:** fungal NTDs, laboratory diagnosis, mycetoma, sporotrichosis, chromoblastomycosis, integrated approaches

## Abstract

The diagnosis of fungal Neglected Tropical Diseases (NTD) is primarily based on initial visual recognition of a suspected case followed by confirmatory laboratory testing, which is often limited to specialized facilities. Although molecular and serodiagnostic tools have advanced, a substantial gap remains between the desirable and the practical in endemic settings. To explore this issue further, we conducted a survey of subject matter experts on the optimal diagnostic methods sufficient to initiate treatment in well-equipped versus basic healthcare settings, as well as optimal sampling methods, for three fungal NTDs: mycetoma, chromoblastomycosis, and sporotrichosis. A survey of 23 centres found consensus on the key role of semi-invasive sampling methods such as biopsy diagnosis as compared with swabs or impression smears, and on the importance of histopathology, direct microscopy, and culture for mycetoma and chromoblastomycosis confirmation in well-equipped laboratories. In basic healthcare settings, direct microscopy combined with clinical signs were reported to be the most useful diagnostic indicators to prompt referral for treatment. The survey identified that the diagnosis of sporotrichosis is the most problematic with poor sensitivity across the most widely available laboratory tests except fungal culture, highlighting the need to improve mycological diagnostic capacity and to develop innovative diagnostic solutions. Fungal microscopy and culture are now recognized as WHO essential diagnostic tests and better training in their application will help improve the situation. For mycetoma and sporotrichosis, in particular, advances in identifying specific marker antigens or genomic sequences may pave the way for new laboratory-based or point-of-care tests, although this is a formidable task given the large number of different organisms that can cause fungal NTDs.

## 1. Introduction

In 2016, the World Health Organisation (WHO) [1] formally recognized mycetoma as a neglected tropical disease (NTD). Mycetoma is a chronic subcutaneous infection caused by over fifteen different species of fungi or filamentous bacteria (actinomycetes) [2,3], and advanced cases often involve bone penetration and relentless destruction and swelling of tissue. Mycetoma can cause substantial disability and in some cases, may prove fatal. In 2017, the equally disfiguring mycosis chromoblastomycosis [4], caused by over 8 different fungal species, was added to the list of WHO-recognized NTDs. This disease causes massive limb swelling, accompanied in some patients by verrucous skin plaques and when left untreated, may cause secondary squamous cell carcinoma. Both mycetoma and chromoblastomycosis are now classified by WHO as fungal NTDs, together with other unspecified deep mycoses. One of these unspecified mycoses meeting many of the criteria for an NTD is sporotrichosis, another subcutaneous fungal infection that shares features with both mycetoma and chromoblastomycosis including comparative neglect and disproportionate impact on impoverished populations. Over the past ten years, sporotrichosis has spread widely in Brazil, to the south and south east of the country from its origins as a zoonotic infection spread from cats in Rio de Janeiro state [5].

Although mycetoma, chromoblastomycosis, and sporotrichosis are different diseases, they share several features. Firstly, they are largely confined to the tropics and affect poor communities, usually in rural areas [6]. In most of the endemic regions, these infections occur sporadically, although in some areas they reach a much higher level of endemicity. For example, the large ongoing Brazilian outbreak is linked to contact with infected cats and largely caused by the species *Sporothrix brasiliensis* [5]. In other parts of the world, foci of infection not associated with cats are more common and these are usually caused by different species such as *S. schenckii* (sensu strictu), or *S. globosa* [7]. Mycetoma occurs worldwide across a region called the global mycetoma belt, which covers tropical countries with low annual rainfall in Africa, Asia, and the Americas. Some localities within these countries report higher incidence of infection (e.g., Sudan and Mexico), although surveillance is limited and true incidence is unknown [2,3]. Mycetoma differs from the other two NTDs in that some endemic areas report a majority of cases caused by actinomycete bacteria rather than fungi, but both bacterial and fungal etiologies result in similarly severe physical and social disabilities including stigma and loss of economic productivity. Strictly speaking eumycetoma caused by fungi is the only true fungal NTD. However, WHO currently classifies both actinomycetoma and eumycetoma under the heading of fungal NTDs or skin NTDs (see below). In chromoblastomycosis hot spots of infection occur in countries with areas of higher annual rainfall including Madagascar, Costa Rica, Brazil, the Dominican Republic, and parts of the state of KwaZulu Natal in South Africa [4]. All three conditions are treated using a process known as innovative and intensified disease management (IDM), which provides antifungal or surgical treatment based on the needs and clinical expression of disease in each individual patient [3]. However, many current antifungals have limited effectiveness and although there are case studies indicating promising activity for some, e.g., voriconazole and posaconazole [8], there are very few formal clinical trials, and because of relative cost, even with the introduction of generic medications, medicines are not available to all who need them. No new medicines for these infections have been introduced in the last 10 years. A new international clinical trial of the antifungal fosravuconazole is underway for mycetoma due to *M.mycetomatis*, but a long journey remains to achieve effective and affordable treatment options.

The responsibility for the initial recognition of these three fungal and skin NTDs often falls on front line health service workers who have received simple basic training. However, the differential diagnosis includes important and complex diseases, such as cutaneous leishmaniasis and mycobacterial infections, as well as tumours and chronic inflammatory dermatoses. Moreover, the appearance of these diseases may be quite different in people living with HIV. Confirmatory laboratory diagnoses depend on specialized medical mycology techniques and expertise. Thus, many cases are probably missed because of a lack of mycological capacity at front line health care centres and peripheral diagnostic laboratories.

All three of these infections present clinically with visible, skin abnormalities [9,10,11], which raises the possibility and underscores the importance of rapid diagnosis. With rapid diagnosis would come faster and more accurate treatment decisions. To begin addressing the need for better and faster diagnosis, a new WHO handbook provides a simple guide for early clinical diagnosis [12]. However, because treatment of fungal NTDs is both lengthy and complex, diagnosis needs to be confirmed in the laboratory. Yet, there is no simple, accurate test akin to a smear for leprosy, card antigen test in lymphatic filariasis and yaws, or molecular identification in Buruli ulcer [11]. Furthermore, consensus is lacking on the best method to confirm fungal NTD diagnoses.

For this reason, a group of clinicians, microbiologists, and mycologists with expertise in the diagnosis of mycetoma, chromoblastomycosis, and sporotrichosis have put forward a consensus on (a) the current optimal or gold standard methods to diagnosis these diseases in well-equipped laboratories and (b) advice on the best methods that can support the diagnosis in resource-limited conditions where access to specialized laboratory facilities is limited. We also provide advice on best practices for obtaining diagnostic specimens.

## 2. Methods

Twenty-six experienced mycologists and clinicians, including dermatologists and infectious disease specialists, recognized for their laboratory or clinical expertise in mycetoma, sporotrichosis, and chromoblastomycosis from different world regions were invited to participate in this survey; of these, 23 responded and completed the assessment. Those who participated included some from endemic regions as well as others from non-endemic zones, but the latter all had extensive practical experience of these diseases in endemic areas. Survey questions included descriptions of current practice in well-equipped diagnostic or clinical settings in each respondent’s country. In addition, questions covered confirmatory tests that could be used in peripheral clinics or laboratories with little expertise in mycological techniques. The best methods of obtaining diagnostic specimens were also discussed

Questions on confirmatory tests included direct microscopy after treatment with 10–20% potassium hydroxide and culture on suitable laboratory media such as Sabouraud’s dextrose agar. Other methods included serology and histopathology with special stains such as periodic acid Schiff reagent or methenamine silver. Molecular diagnostic tests were also included, such as probe-based Polymerase Chain Reaction (PCR) techniques, sequencing, and Matrix Assisted Laser Desorption/Ionisation Time of Flight (MALDI-ToF). In addition, participants were asked to indicate where other diagnostic or assessment tools were useful to support the management of patients. These included imaging methods such as Magnetic Resonance Imaging (MRI), ultrasound and other point-of-care tests such as dermoscopy (in situ microscopy of the skin).

Results were collated and presented to members of the group in a workshop where the choices of diagnostic methods were clarified and refined further to reach an overall consensus

## 3. Results

### 3.1. Well-Equipped Clinical Centres and Laboratories

There was significant agreement on the optimal Fungal NTD diagnostic methods in the well-equipped settings. For instance, >83% of respondents agreed on the use of clinical features and culture mycetoma, chromoblastomycosis, and sporotrichosis (Table 1).

For mycetoma > 88% recommended the combined use of direct microscopy (Figure 1a,b), culture and histopathology (Figure 2a,b) together with clinical evaluation. While respondents recognized the increasing importance of molecular diagnostic tests, such as specific probe-based PCR and sequencing, these are not yet available in all settings. There was also a high degree of support for the use of imaging, MRI, X-ray, or ultrasound as important adjuncts to diagnosis of mycetoma particularly for planning management strategies.

There was similar agreement in chromoblastomycosis where over 88% recommended the use of direct microscopy (Figure 3a), histopathology (Figure 3b), and culture with clinical features. There was less support for the use of molecular diagnostic tests (50%) but their use is evolving. Although dermoscopy has not been widely used in the diagnosis of chromoblastomycosis, some advocated for its value as a point of care test.

With Sporotrichosis, responses were more specific, with culture as the most favored diagnostic option (96%) together with some clinical features; most respondents specified lymphangitic spread as a hallmark clinical feature. Half of respondents advocated for the use of molecular tests for the diagnosis of sporotrichosis and species-level identification. Certain *Sporothrix* species have unique epidemiologic characteristics, and thus unique and public health implications and species-level identification can only be achieved by molecular methods. However, the frequent absence of organisms detectable either by direct microscopy or histopathology presents a major challenge to sporotrichosis laboratory confirmation. In discussion, some expressed the view that microscopy and histopathology were useful in order to exclude clinically similar infections such as cutaneous leishmaniasis, but all agreed that these two laboratory procedures were ancillary and could not be recommended for primary diagnosis. Imaging is sometimes useful in widespread Sporotrichosis to exclude internal dissemination.

### 3.2. Peripheral Clinics and Laboratories

When asked about the use of diagnostic procedures in resource-limited field settings where access to a well-equipped laboratory was unlikely, responses were different. Nonetheless, there was considerable agreement on a minimum diagnostic set that includes clinical feature evaluation (Table 2). All recognized that culture-based diagnosis was seldom available away from main centres.

For mycetoma, clinical features (96%) coupled with direct microscopy (88%) were favored by respondents as diagnostics that should trigger referral to a well-equipped reference laboratory. Similar agreement on the value of these same two methods was also reported for chromoblastomycosis. Experts disagreed on the use of histopathology for both mycetoma and chromoblastomycosis. The crux of this disagreement stemmed from the recognition that capacity for histopathological diagnosis differs widely in certain areas, including in resource-limited setting [13]. Ultimately, all agreed that, where capacity exists, histopathology is an important primary diagnostic test.

Results differed for sporotrichosis. At least half of respondents reported that the simplest of laboratory tests available at peripheral levels (i.e., direct microscopy) are unhelpful for sporotrichosis diagnosis, and that histopathology should not be prioritized. As reported for well-equipped settings, respondents felt that direct microscopy and histopathology were primarily useful for excluding similar clinical syndromes such as cutaneous leishmaniasis and not for positively identifying sporotrichosis. For direct microscopy, the requirement of special stains (e.g., Giemsa stain) to identify and rule out *Leishmaniasis* as a diagnosis is an additional obstacle for small clinics. Therefore, these tests are not helpful in guiding referrals to expert centres and were not recommended for front line sporotrichosis diagnosis. Sample collection methods (Table 3).

For most participants, superficial sampling, such as swabbing, or impression smears, were not useful for the diagnosis of these three NTDs. Removing tissue material by scraping the surface where it is broken and moist, or preferably, by incision or excision and deep biopsy or curettage were the preferred methods among 58–79% of the respondents. The majority (67%) support for skin scraping in chromoblastomycosis reflects the fact that, provided that the correct site (where small black dots can be seen on the skin surface) is selected, the characteristic pigmented “muriform” cells of chromoblastomycosis can be seen with direct microscopy in skin surface scrapings, as the fungal cells are eliminated trans-epidermally. On discussion it was agreed that the best site to obtain these is the surface of the lesion where there are dark areas (black spots), that indicate where clusters of pigmented cells are present. For sporotrichosis, *Sporothrix species* are detected in culture from skin lesion samples in the first 7 to 8 days of incubation although most (78%) supported the use of material taken by curettage or biopsy. Taking deeper tissue samples may be a rate-limiting step in accessing laboratory supported diagnosis in resource-limited environments as this process requires equipment such biopsy punches or curettes, together with local anesthetic, which are often not available in local primary care settings.

## 4. Discussion

Our survey and subsequent discussions identified areas of strong agreement on the best practices for confirming fungal NTD diagnoses. This consensus provides a framework for improving diagnostic selection, defining referral pathways, and improving access to laboratories with the appropriate capacity. The disparity between recommendations for well-equipped and resource-limited settings was telling. Where experts agreed on use of “older” diagnostic methods such as culture and microscopy, the primary rational was based on practicality and available resources. Survey respondents acknowledged the substantial advances in new diagnostic methods for fungal NTDs and their increasingly important roles in typing and speciation, in spite of limited access to date in endemic areas.

Concerted efforts have been made to improve field diagnosis in remote settings through comprehensive and simple training methods, largely based on visual clues to diagnosis [12]. But adapting or developing new diagnostic tests, drawing on the results of recent research was identified as a critical need for the future. For mycetoma, important recent developments with direct impact on diagnosis include molecular methods to identify new etiologic agents, isothermal amplification techniques, the adaptation of MALDI Tof to fungal identification and the finding of novel infection-related antigens [14,15,16,17]. A new grading system for Magnetic Resonance Imaging (MRI) allows clinicians to assess extent and severity of mycetoma, which may not be apparent on visual inspection [18]. The challenge of adapting new tests to in endemic areas are considerable, but these modalities are sorely needed. A recent survey of communities in an endemic zone for mycetoma in Sudan shows the benefits of active case finding, but also the non-specific nature of current diagnostic definitions based on clinical signs and symptoms, used at community level, which leads to the identification of many false positive cases [19]. The dermatoscopic features and diagnostic utility of pale grain [20] and dark grain mycetomas have been described [21] and with validation, may prove a viable point-of-care diagnostic indicator. Although dermoscopy was not a component of our survey, it appeared repeatedly in discussion. A major obstacle to wider use of dermoscopy is the required training and cost of the dermatoscope. These obstacles have been overcome for chromoblastomycosis diagnostics with the use of molecular and MALDI ToF analyses [22,23,24]. Chromoblastomycosis may present in a number of morphological forms ranging from shallow plaques, to verrucous plaques, sinuses, and ringworm-like lesions, and improving the accuracy of diagnosis is a key target [25,26]. Again, dermoscopy is a possible point of care option in settings where there are health care workers trained in its use. However, the dermatoscopic features of chromoblastomycosis are not well recognized among those working in this field. For example, while clusters of golden/brown structures in the skin are indicative of pigmented fungal cells [27], other features are non-specific, and distinguishing useful features must be learned.

The clinical presentations of sporotrichosis are notoriously pleomorphic which presents an obstacle to translation into usable visual clues for diagnosis, for use at field level, and some cases are complicated by additional distractors such as secondary immunological reactions, erythema nodosum, and erythema multiforme [28,29]. In addition, sporotrichosis presents a diagnostic challenge, although more accurate molecular tools have begun to change our understanding of the importance of identifying the newly identified species in terms of transmission and treatment strategies [7,30,31,32]. There have also been major advances in identifying antigens associated with infection [33]. An ELISA assay has been used in one retrospective study of patients with sporotrichosis and yielded promising results [34]; serological detection is associated with at least two specific antigens in the cell wall of *Sporothrix* species [35] and building on this work may provide new options for diagnosis. There is also a commercial latex agglutination test for the detection of antibodies to *Sporothrix*, although in cutaneous disease antibody titres are not high and false positive reactions at low titre have also been recorded. Nonetheless, there are formidable difficulties in accurately recognizing sporotrichosis in peripheral settings, as the hallmark clinical features of Sporotrichosis such as lymphangitic spread are common to other pathogens, such as *Leishmania*, which often share the same endemic areas [36]. Direct microscopic examination or histopathology of cases of sporotrichosis are of often of little value as there are usually very few organisms seen, although the presence of a refractile eosinophilic fringe or asteroid body in sporotrichosis, where it can be found, is a very useful laboratory diagnostic sign. Culture remains the most used laboratory diagnostic test where this is available. Screening suspected cases with an intradermal skin test antigen test is one possible diagnostic option that could be applied in peripheral settings [37], but the reagents are not widely available and skin reactions may also be positive in exposed but uninfected individuals. The dermatoscopic appearances of sporotrichosis have also been reported [38] but similar dermatoscopic appearances are also seen in other systemic mycoses disseminating to the skin such as cryptococcosis or histoplasmosis which limits its diagnostic use [39]. Accurate laboratory diagnostic tests are available for these latter two systemic mycoses. From a practical standpoint those diseases that may manifest a lymphangitic pattern of spread such as sporotrichosis, cutaneous leishmaniasis, non-tuberculous mycobacterial infection, and nocardiosis all require special treatment. As an indicator for referral to a specialized facility, a chain of skin lumps suggesting lymphangitic spread, as a clinical sign, is worth considering as part of a diagnostic algorithm. However, it does not address confirmation of the diagnosis in other clinical presentations of sporotrichosis.

Survey respondents were selected for their expertise in both clinical and laboratory diagnostic approaches to fungal NTDs, and many also have training and experience in well-equipped settings in developed countries. Often, however, clinicians and laboratories in endemic areas are not familiar with these procedures. Likewise, they have little access to newer diagnostic techniques including molecular diagnostics. Experience and competency is often limited, in technical clinical procedures such as performing a skin biopsy safely with local anesthetic and in diagnostic techniques such as direct microscopy and histopathology of fungal disease. As a start to mitigating the impact of limited resources on capacity, a free online training module has been launched recently by the Fungal Infection Trust (www.microfungi.net), and may contribute to improving such skills, but there remain large gaps. The World Health Organisation, together with subject specialists, is in the process of addressing these needs through the promotion of a skin NTD initiative [40], which specifically includes the fungal infections.

Although there are some examples of new laboratory tests that may simplify the diagnostic journey of the patient presenting with the fungal NTDs mycetoma, chromoblastomycosis and sporotrichosis, there is a real and urgent need for innovation in diagnostics. The development of accurate antigen detection tests for any of these diseases would dramatically change diagnostic access and performance, particularly if available for use at the point of care. For mycetoma specifically, a field-ready serologic assay that can distinguish the fungal variant of mycetoma from the visually similar bacterial variant could prevent patients from receiving incorrect and ineffective treatments. However, diagnosis to species level would require a formidable array of different tests given the large number of different species that cause mycetoma. A similar qualification applies to chromoblastomycosis. The modification of molecular techniques for use at the peripheral laboratory level is a second option with great potential impact subject to the same concerns about the large number of different potential causes. For now, small steps are also worth the effort [41]. For example, supplementing simple visual diagnostic methods [12] with information key characteristics such as lymphangitic spread of skin lesions could improve specificity in a clinically relevant way. In addition, a simple training guide on direct microscopy in small hospital settings would improve diagnostic capacity for mycetoma and chromoblastomycosis. For many years, mycologists have lamented the limited availability of simple, accurate and specific laboratory tests, yet almost all attention has focused on the invasive infections of the severely immunocompromised, such as invasive aspergillosis. However, there is an urgent need to improve existing technologies for the diagnosis of the fungal NTDs seen in resource poor settings, in order to reduce the burden of these debilitating diseases.

## Figures and Tables

**Figure 1 tropicalmed-04-00122-f001:**
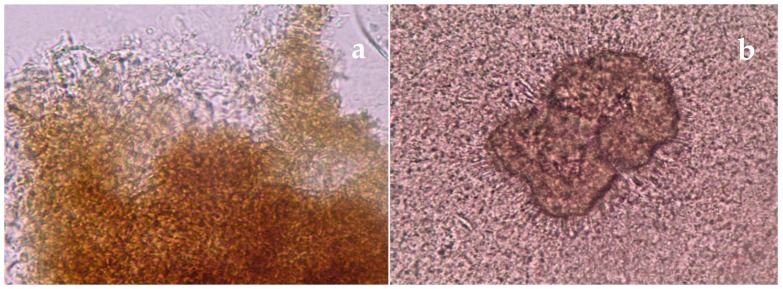
(**a**) Direct microscopy (15%) potassium hydroxide (KOH)). Dark grain eumycetoma (fungal mycetoma) × 40; (**b**) Direct microscopy (15% KOH). Pale grain actinomycetoma due to *Nocardia brasiliensis* × 40.

**Figure 2 tropicalmed-04-00122-f002:**
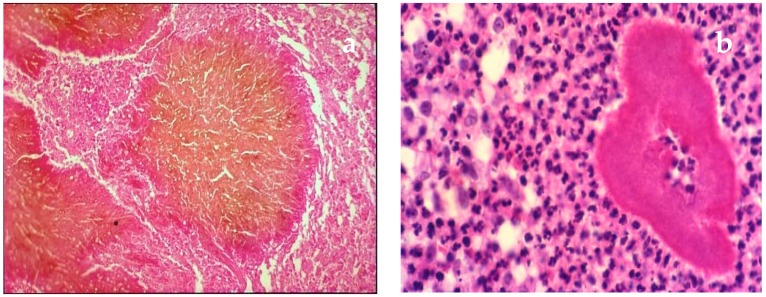
(**a**) Surgical biopsy. Dark grain of eumycetoma M.mycetomatis (Haematoxylin eosin HE) × 40; (**b**) Deep biopsy. Grain of *Nocardia brasiliensis* (HE) × 40.

**Figure 3 tropicalmed-04-00122-f003:**
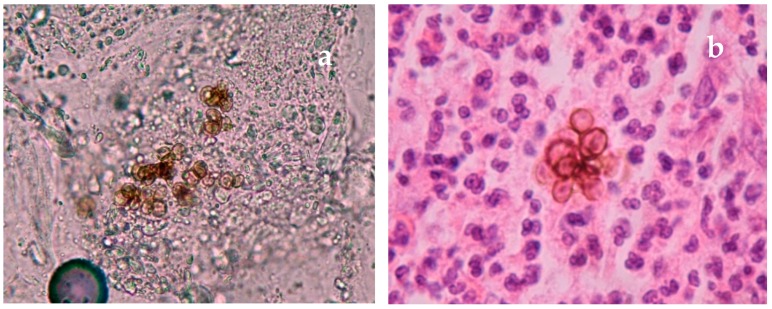
(**a**) Direct microscopy skin scales (15% KOH). Muriform cells typical of Chromoblastomycosis; (**b**) Skin biopsy. Muriform cells of chromoblastomycosis (HE) × 40.

**Table 1 tropicalmed-04-00122-t001:** Optimal diagnostic methods in a well provided laboratory—positive responses.

Disease	Clinical Features	Direct Microscopy	Culture	Serology	Molecular Diagnosis	Histopathology	Other
Mycetoma	92%	88%	96%	8%	71%	88%	imaging, dermoscopy
Chromoblastomycosis	88%	92%	96%	8%	50%	92%	dermoscopy
Sporotrichosis	83%	25% ^1^	96%	4%	50%	67%	intradermal test

^1^ The importance of microscopic features in Sporotrichosis diagnosis elicited a qualified response (see below).

**Table 2 tropicalmed-04-00122-t002:** Diagnostic methods of use in peripheral clinics and laboratories (positive results).

Disease	Clinical Features	Direct Microscopy	Culture	Serology	Molecular Diagnosis	Histopathology	Other
Mycetoma	96%	88%	33%	-	13%	43%	imaging
Chromoblastomycosis	96%	92%	33%	8%	4%	54%	
Sporotrichosis	88%	42%	50%	4%	45	21%	

**Table 3 tropicalmed-04-00122-t003:** Method of collecting material for laboratory investigation (positive responses).

Responders Choice	Swab from Broken Skin or Sinuses	Impression Smear	Skin Scraping from Broken Skin or Sinuses	Punch or Incision Biopsy or Curettage	Excision Biopsy
Mycetoma	38%	42%	54%	74%	61%
Chromoblastomycosis	8%	29%	67%	79%	42%
Sporotrichosis	21%	17%	50%	75%	58%

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
