# Peer review of "The Diagnosis of Fungal Neglected Tropical Diseases (Fungal NTDs) and the Role of Investigation and Laboratory Tests: An Expert Consensus Report"

_tropicalmed, 2019, doi:10.3390/tropicalmed4040122_

Round 1

Reviewer 1 Report

The authors have presented an ‘Expert Consensus Report’ on diagnosing Fungal Neglected Tropical Diseases. Like all consensus documents it lacks some of the clarity and authority of a more considered review and there are parts that give a very misleading impression of the diagnostic value of various approaches. However, with modification it could provide a useful starting point for developing diagnostic guidelines.

I have noted particular sections that could do with a bit more explanation and have also noted some minor typographical errors, which are presented below in the order that they appear in the text.

Line 81: suggest replace ‘gene’ with ‘fungal genomic sequences’ Line 88: Surely a major distinguishing feature of chromoblastomycosis is the distinctive papillomatous lesions? Line 89: Suggest inserting …can cause secondary squamous cell carcinoma. Line 166: ….clinical features and culture for mycetoma…. Line 230 Table 3: I was very surprised to see that 38% of respondents supported a swab sample for the diagnosis of mycetoma and feel that this needs some qualification such as ‘swab of a discharging sinus’. A surface swab alone would be of very limited value for all these conditions. Likewise a skin scraping would be of very limited value in mycetoma where the grains containing the fungal mycelium are usually deep within the tissues. The authors do not want to appear to be advocating sampling methods with little intrinsic value in these conditions as negative results may lead an inexperienced clinician to dismiss the possibility of such a condition. Line 263 delete ‘to’ Line 272: change of font half way through sentence Line 297: there should be some mention of the typical asteroid bodies that may be seen on histological diagnosis of sporotrichosis. Line 303: Although sporotrichosis may resemble the tissue dissemination of cryptococcosis this disease is now very easily diagnosed by application of the lateral flow device which can even be used on finger-prick blood so cryptococcosis should be easy to exclude from the differential diagnosis even in resource-limited environments. Likewise there are serological tests for histoplasmosis (antibody or antigen) and a lateral flow device under development to help exclude this diagnosis. Line 322: The value of an antigen detection test should not be overplayed for mycetoma where there exist a very broad range of agents causing both pale grain and dark grain eumycetoma so it would have to be a multiplex test. There are also several different fungal genera consistently implicated in chromoblastomycosis. Line 324: This is a massive oversimplification as the ‘fungal variant’ of mycetoma includes numerous different genera and species – this needs some discussion as a single serological test is unlikely to be helpful. Moreover there are often marked differences in the susceptibility of the different causative agents to antifungal drugs so knowing which organism is causing disease can be helpful in the management strategy. This will only be achieved through culture or molecular diagnosis to species level. In a resource limited setting it may be possible to use an FTA filter disc (Whatman) to extract fungal DNA from grains or biopsy specimens and then post them off for genomic analysis.

Author Response

I have attached our answers to referee 1 in the attached file

Reviewer 2 Report

Title: The diagnosis of Fungal Neglected Tropical Diseases (Fungal NTDs) and
the role of investigation and laboratory tests: An expert consensus report
Journal: Tropical Medicine and Infectious Disease

The manuscript by Hay et al purports to be an expert consensus review on the role of various investigational and laboratory tests to aid in the diagnosis of three fungal neglected tropical diseases (NTDs): eumycetoma, chromoblastomycosis and sporotrichosis. Fungal NTDs are an important area of medical mycology, and in this respect such a report should be an important addition to the literature. Unfortunately, in its current form, this report does not fulfil that expectation. There are a number of issues, many stemming from inaccurate generalisations, that require addressing before the manuscript could be considered for publication.

Lines 80-81 and elsewhere: the identification of marker antigens for eumycetoma are unlikely to easily pave the way for new laboratory tests. Whilst such an approach might very well be useful in areas where a single eumycetoma agent predominates (eg M. mycetomatis in Sudan), the number of different eumycetoma agents described from cases in S. America and the Indian subcontinent in particular would require th development of a very extensive panel of disparate antigens for such an approach to have utility. As the authors discuss, mycetoma can be caused by fungi or actinomycetes. Since this report concerns fungal NTDs, I suggest the authors use eumycetoma throughout to clarify the distinction. The English requires improvement in several sections (eg lines 73, 165, 198) Lines 89-90 are a repetition of the previous section which had already explained that mycetoma and chromoblastomycosis have previously been recognised as NTDs by WHO. Line 114: space and punctuation missing. Lines 114-116: I would disagree with this generalisation regarding the effectiveness (or lack thereof) of currently available antifungal agents. Several of the currently available antifungal agents are highly effective (particularly voriconazole, which is now off-patent). The lack of effectiveness is principally due to the advanced degree of disease progression at the time of presentation, when antifungal therapy alone (with any antifungal agent) in the absence of surgery is highly unlikely to be effective. Lines 116-117: to my knowledge, this international trial is only aimed towards patients with molecularly-proven infection with M. mycetomatis. See the point above (point 1) concerning the ever-increasing list of novel agents of eumycetoma. Figures 1a and 1b and 2a are of poor quality (at least in the pdf version available for review). Figure 2b – is this really useful as the report concerns fungal NTDs? Table 3: There are several issues with this Table that again stem from over-simplification. Swab or impression smear sampling can be extremely useful for the diagnosis of eumycetoma if there are actively draining sinuses with fungal grains present in exudate. In the absence of sinuses/grains these techniques are of absolutely zero value. Were such distinctions included in the survey? They should at least be discussed in the text. There appears to be an issue with the font on lines 272-273. Finally, I am unsure of the real value of the percent scoring system used in tables 1-3, especially when no distinction is made between responses from resource-limited settings where eumycetoma is endemic and resource-rich settings. Perhaps of more utility would be an additional Table that summarises the techniques and tests that are available in centralised as opposed to rural/local laboratories in the resource-limited areas where the disease is endemic, since a proportion of the invited experts at best see sporadic, imported cases. Although according to the text 23 of 26 invited experts responded to the survey, only 22 authors are listed?

Author Response

I have added our responses to Referee 2 in the attached file

Reviewer 3 Report

This is a very good study presenting a consensus report of diagnostic of fungal neglected tropical diseases in low settings and well-equipped laboratories. These fungal NTDs are amongst the most neglected and this is a very important study in this field but for NTDs in general.

I recommend this paper for publication however the format and presentation of the entire manuscript really need to be checked accurately for consistency. It looks really sloppy.

Here are some specific comments.

Abstract

L 78-79: No need to capital letters at essential diagnostic tests

Introduction

Lack of references at the beginning. Reconsider the position of your references at the end of the sentences rather than in the middle.

L 92: Remove “,” after features.

L109: Ref?

Results

The first headline on well-equipped laboratories should be 3.1 and format adjusted to match the rest of the text. Following headlines should be adjusted accordingly. And modify tittle for well-equipped instead of well equipped as this is how it is written everywhere else in the text.

L 164: fungal NTD without capital F, check this through the text as you sometimes put and F and other time not.

L 165: It rather seems to be >83%

L 166: For mycetoma… It seems that a word is missing in your sentence.

L 181: Wy : after x40 (see similar comment for other figures).

L 184: No capital G at grain.

L 185: Be consistent, you used >85% before and then you use here over 85%; Also it’s not really clear when we read the text what it is referring to in the tables as numbers don’t match as it seems here that numbers are 92%, 92% and 96%.

L 187: Avoid repeating although.

L 191: Figure 1c should be after 1b and not after 2b. If this doesn’t fit with your text then consider changing the numbering of your figures.

L 212: By over 88%, use exact numbers.

L 230: Why ? after positive responses

L 233: It seems that it’s rather 54-79% that 58-80. Again, you have the exact % of answers so use the exact results.

L 242-243: This whole sentence doesn’t make sense to me. Where this result on Sporotrichosis detection come from? This doesn’t seem appropriate in the sample collection method section. Is it something observed by the authors or from reference which in the last case is not indicated and again don’t seem at the right place here.

Discussion

L247: No capital F at fungus (also in line 252)

L 272-273: check format

L 284-287: This is quite heavy, consider rephrasing this. The section “…revolutionise our understanding of the importance of identifying…” could maybe be alleviated a bit. Also mention a diagnostic challenge without detailing further here.

Further discussion on differences between low and well-equipped settings was expected based on your abstract and results. Can you develop this further in your discussion.

Check your references for consistency. Some have the direct link whereas others don’t. is this something requested by the journal?

Check all your text for presence of double space all along the text but also for the absence of space in some sentences…

Author Response

I have added our responses to Referee 3 in the attached file

Round 2

Reviewer 2 Report

The authors have satisfactorily addressed my concerns with the original version of this manuscript